# *Salmonella* Prophages, Their Propagation, Host Specificity and Antimicrobial Resistance Gene Transduction

**DOI:** 10.3390/antibiotics12030595

**Published:** 2023-03-16

**Authors:** Lisa Trofeit, Elisabeth Sattler, Johannes Künz, Friederike Hilbert

**Affiliations:** Institute of Food Safety, University of Veterinary Medicine, 1210 Vienna, Austria

**Keywords:** *S. enterica* serovar Enteritidis, *S. enterica* serovar Typhimurium, resistance, prophage, phage therapy, foodborne pathogen

## Abstract

*Salmonella enterica* subsp. *enterica* is a zoonotic bacterial pathogen that causes foodborne outbreaks in humans. Lytic bacteriophages to control *Salmonella* in food production are already being used in scientific studies and some are commercially available. However, phage application is still controversial. In addition to virulent phages, which are used in phage therapy and lyse the bacterial host, lysogenic phages coexist in the environment and can reside as prophages in the bacterial host. Therefore, information about *Salmonella* prophages is essential to understand successful phage therapy. In 100 *Salmonella* food isolates of the serovars Enteritidis and Typhimurium, we propagated prophages by oxidative stress. In isolates of the serovars Typhimurium and Enteritidis, 80% and 8% prophages could be activated, respectively. In the phage lysates from the serovar Typhimurium, the following antibiotic resistance genes or gene fragments were detected by PCR: *sul1*, *sul2*, *bla_TEM_*, *strA* and *cmlA*; however, no *tetA*,*B*,*C*, *bla_OXA_*, *bla_CMY_*, *aadA1*, *dfr1*,*2* or *cat* were detected. In contrast, no resistance genes were amplified in the phage lysates of the serovar Enteritidis. None of the phage lysates was able to transduce phenotypic resistance to WT 14028s. Most of the prophage lysates isolated were able to infect the various *Salmonella* serovars tested. The high abundance of prophages in the genome of the serovar Typhimurium may counteract phage therapy through phage resistance and the development of hybrid phages.

## 1. Introduction

Salmonella (S.) enterica is an important zoonotic bacterial pathogen transmitted to humans through food, mainly through contaminated poultry products and eggs [1]. *S. enterica* subsp. *enterica*, with its main serovars, Enteritidis and Typhimurium, is the second most important bacterial zoonotic pathogen causing foodborne outbreaks in humans in Europe [1]. Poultry meat, eggs and food thereof are the main sources, but other animal-based foods and plant products have also been implicated in foodborne outbreaks of Salmonella [1]. A good proportion of these Salmonella isolates are multi-resistant to various antibiotics [2,3,4,5,6,7,8,9]. When nontyphoid Salmonella causes invasive clinical signs of disease in humans, antibiotic treatment is required. Invasive disease occurs primarily in the YOPI group (Young, Old, Pregnant, Immune Deficient) of humans [10,11,12]. As a facultative intracellular pathogen, Salmonella can multiply in macrophages and produce endotoxins during infection. Certain antibiotics may enhance intracellular location and endotoxin release, which limits the range of treatment options [13,14].

Lytic bacteriophage to control Salmonella in food production and to combat animal colonization and disease are used intensively in scientific studies and some are already commercially available [15,16,17]. Phage therapy in humans still requires individual approval in most countries and results of phage application are still controversial. Reports of phage therapy for human salmonellosis are scarce [18].

Virulent phages are mainly used in therapy, but lysogenic phages, which can enter the bacterial genome and survive as a prophage for a period of time, appear to be most relevant to bacterial evolution. Horizontal gene transfer by phages is called transduction and refers to lysogenic phages, but even virulent phages are capable of advertently transducing genetic material from one bacterial cell to another [19]. Therefore, the fitness of bacteria can be enhanced by prophages harbouring phage morons (adding “more on” to the bacterial host), which are genes carried by the prophage and benefit the bacterial host [20]. These accessory genes can supply virulence factors, toxins and resistance genes. Furthermore, the infection of a bacterial cell by virulent and lysogenic phages can lead to so-called hybrid phages, which can change in host specificity or live style (virulent/lysogenic). In this way, initial information about the phageome of a specific bacterial species or genus can be used to secure phage therapy.

Here we analyzed the *S*. serovars Typhimurium and Enteritidis for inducible prophages. In addition, these phage lysates were screened for antimicrobial resistance genes (morons) and the transduction of phenotypic resistance to a standard strain. Finally, we evaluated the host specificity of these phage lysates.

## 2. Results

Food isolates of the *S.* serovars Typhimurium and Enteritidis, for the propagation of prophages, were obtained from different types of meat purchased in Austria, and human isolates were provided by the Austrian National Reference Center for Salmonellosis. As host strains to culture the activated prophages from the food and human isolates, we selected two strains from the type and culture collections (ATCC and DSM) previously described in the literature as well-defined hosts. These culture collection strains have already been used for phage transduction experiments, and do not harbor inducible prophages in their genome to interfere with phage lysates from human and food isolates. The strain ATCC 14028 WT was selected as a known strain for use as a host strain for the provision of transduction libraries, and has also been used in phage infection and transduction studies [21]. The strain DSM 17420 has also been used for transduction studies [22]. Various stressors can activate prophage induction. We tested H_2_O_2_, nitrate and mitomycin C in a study on Sty1, a serovar Typhimurium isolate, and WS27D, a serovar Enteritidis isolate, respectively. Mitomycin C, a cytostatic agent, was most effective at inducing prophage, followed by H_2_O_2_ and nitrite. Therefore, for prophage induction, H_2_O_2_ was used as a potential oxygen stressor for *Salmonella* (a typical activated prophage infecting host strains is shown in Figure 1). In Figure 1, a typical propagated prophage lysate was used to infect the host strains ATCC 14028 (left) and DSM 17420 (right). In the overlay method, an exponentially growing host strain at a McF concentration of 0.5 is used with a phage extract (in this case a propagated prophage lysate from a *S*. serovar Typhimurium isolate, left, and a *S*. serovar Enteritidis isolate, right, with H_2_O_2_), which is sterile filtered to avoid bacterial contamination and treated with DNAse, avoiding interaction with free DNA.

Additionally, we tested four isolates each of the serovar Typhimurium and the serovar Enteritidis, treated in parallel with H_2_O_2_ and mitomycin C, respectively. H_2_O_2_ was able to propagate prophages in two isolates of the serovar Typhimurium, whereas mitomycin C was only able to induce it in one of the isolates. No difference was seen for activating prophage with either H_2_O_2_ or mitomycin C in the four isolates of the serovar Enteritidis.

In 40 out of 50 *S*. serovar Typhimurium food and human isolates, prophages were inducible by H_2_O_2_ treatment. There was no difference between the human and food isolates in prophage activation potential under H_2_O_2_ treatment. In a total of 20% of the serovar Typhimurium isolates, no phage transduction was possible under H_2_O_2_ treatment and host strain ATCC 14028 WT. The activated prophage lysates of these isolates were also unable to infect DSM 17420. For the serovar Enteritidis we used DSM 17420 and ATCC14028 WT as host strains for the transduction experiments. Prophages could be activated in only four isolates. All four isolates were isolated from chicken meat. No prophages from any human *S*. serovar Enteritidis isolates could be activated by H_2_O_2_ treatment.

Phages are important evolutionary drivers in bacteria. Antimicrobial resistance genes can be transduced by a variety of lysogenic phages. The potential of phages for horizontal gene transfer, including antibiotic resistance genes, has only recently been discussed [23]. Therefore, we isolated DNA from the total lysates of activated prophages and analyzed it by PCR for 14 common antibiotic resistance genes. We made sure only phage DNA was extracted by filtering lysates and DNAse treatment to destroy external DNA. Of these genes, *sul1* was detected in 5 prophage lysates, *sul2* in 8 lysates, *bla_TEM_* in one lysate, *strA* in four lysates and *cmlA* in 10 isolates (as shown in Table 1). No lysate tested positive for *tetA,B,C, bla_OXA_*, *bla_CMY_*, *aadA1*, *dfr1*,*2* or *cat*. All the lysates that tested positive were from the *S.* serovar Typhimurium isolates and none was from the serovar Enteritidis. Most of the isolates tested positive for one antibiotic resistance gene fragment, but four lysates tested positive for two and two for three genes. A combination of more than one resistance gene, namely *sul1* + *sul2* + *strA* in two isolates, *sul2* + *strA* in two isolates, *cmlA* + *sul1* in one, and *bla_TEM_* + *sul2* in one isolate, was detected.

Transduction experiments were performed in ATCC 14028 WT from each of the lysate described in Table 1, and screening for antibiotic resistant transductans was conducted. Neither lysate was able to transduce phenotypic resistance into ATCC 14028 WT. ATCC 14028 does not harbor any of the resistance genes analyzed and has no phenotypic resistance against the tested antibiotic substances. No growth on plates with adequate antibiotic concentration (ampicillin at 15 µg/mL, sulfonamids at 25 µg/mL, streptomycin at 15 µg/mL and chloramphenicol at 30 µg/mL) was detected.

In order to test the lysing ability of all the different propagated prophages, a number of different *Salmonella* serovars were tested by spot lyses (as shown in Figure 2).

Most prophage lysates of the serovar Typhimurium were able to infect food isolates of the serovar Typhimurium, serovar Enteritidis, serovar Heidelberg, serovar Blockley, serovar Hadar, serovar Rissen, serovar Kentucky and serovar Parathyphi. Only two lysates could infect the serovars Blockley and Paratyphi, whereas the serovars Virchow and Heidelberg were most susceptible and could be infected by 39 and 36 of the total of 44 lysates, respectively (see Table 2). All the data on lysing ability can be found in Appendix A.

In addition, 50 *E. coli* food isolates and 50 *Yersinia enterocolitica* food isolates were tested for susceptibility to the activated prophage lysates. None of these isolates could be infected by any of the 44 lysates extracted from *S*. serovar Typhimurium and serovar Enteritidis (see Appendix A).

## 3. Discussion

The *S.* serovars Typhimurium and Enteritidis are the major foodborne pathogens causing foodborne outbreaks worldwide. Depending on the serovar, this pathogen can be highly resistant to several antimicrobial agents. Therefore, alternative treatment options such as phage therapy are being discussed. *Salmonella* phage applications are already used in many countries in the food sector to avoid food contamination. Phage therapy in animal production is also reaching the stage of large-scale industrial trials [24,25,26,27]. However, these applications can be controversial, as phages drive bacterial evolution in *Salmonella* in the fastest and most effective way [28]. Some *S*. serovars such as Typhimurium are known to harbor functional and truncated prophages in their genome. Some of these prophages can confer phage resistance, while others can interact with phages infecting the cell and fuse into so-called hybrid phages with a different host specificity or lifestyle (virulent or lysogenic), and the transfer of antibiotic resistance genes is also possible [23].

To keep phage application and therapy effective, information about naturally occurring prophages in different serovars is as important as infectivity and host specificity. Here, we propagated prophage from the two most important clinical *S*. serovars, Typhimurium and Enteritidis, and tested their host specificity and their potential to transduce antimicrobial resistance genes.

The most potent activation of prophages in vitro is activation by mitomycin C, a cytostatic agent with potent mutagenic activity. While *Salmonella* isolates are unlikely to come into contact with mitomycin C in their lifecycle, we used oxidative stress from H_2_O_2_ to activate prophage induction, a stressor that *Salmonella* is exposed to in the animal and human host, e.g., against innate immune response, and externally by reactive oxygen species used in food production, such as nitrate, nitrite or in the environment by UV light [29]. Oxidative stress with H_2_O_2_ was almost as effective as mitomycin C in propagating prophages of the *S*. serovars Typhimurium and Enteritidis [30]. In our study, we also tested four isolates of the serovar Typhimurium and the serovar Enteritidis, each treated in parallel with H_2_O_2_ and mitomycin C, respectively. In two of the four serovar Typhimurium isolates, H_2_O_2_ was able to propagate prophages, whereas mitomycin C was only able to propagate prophages in one isolate. No difference was seen in the serovar Enteritidis isolates. Thus, there is a difference in the ability to propagate prophages with different methods, and genome analysis alone is not able to predict the propagation of complete prophages by various stressors.

Interestingly, a large difference was found between the serovars Typhimurium and Enteritidis in prophage propagation. In case of the serovar Typhimurium, 80% of the isolates could be propagated and only 8% of the isolates of the serovar Enteritidis. All four of the isolates from the serovar Enteritidis that could be propagated were food isolates, while the same proportion of human and food isolates of the serovar Typhimurium could be propagated. This is consistent with other studies looking at prophages in *Salmonella* [31,32]. These results confirm the high plasticity of the *S*. serovar Typhimurium genome.

We did not analyze the lysates for reinfection properties, as we did not know if the propagated lysates harbored one or more phages. It can be seen from published genome sequences that many sequenced *Salmonella* genomes hold more than one intact prophage and it is not possible to isolate different phages based on plaque morphology. Additionally, isolation of a single phage from a plaque plate is not always possible. Thus, reinfection studies would have been less meaningful for the definition of phage resistance or superinfection.

Antibiotic resistance genes detected in the activated prophage lysates of the *S*. serovar Typhimurium are also a sign of this plasticity. Gene coding for sulfonamide resistance or a gene fragment was detected most frequently in our study. The two resistance genes *sul1* and *sul2* were identified in two lysates. A combination of sulfonamide resistance genes with the streptomycin resistance gene *strA-B* was found in four lysates. The detection of streptomycin resistance genes in the lysates was always found in combination with another resistance gene or resistance gene fragment. The chloramphenicol resistance gene *cmlA* was detected in 10 lysates, while another chloramphenicol gene, *cat*, was not found in our lysates. A ß-lactam antibiotic resistance gene, the *bla_TEM_*, was identified in only one isolate. No resistance gene to tetracycline could be detected, which is a common resistance phenotype in various *S*. serovars [33]. This could indicate that horizontal transmission of the most common resistance traits, such as ß-lactam resistance and tetracycline resistance, occurs not through phages and transduction events, but through other mobile elements such as plasmids and transposons.

Most phages are very host specific, and broad host range phages are the exception. Regarding other genera, no propagated prophage lysate in our study was able to infect food isolates of *E. coli* or *Y. enterocolitica*. In contrast, most propagated prophage lysates were able to lyse a wide range of different *S.* serovars. No *S*. isolate of other serovars was tested in our study that could not be lysed by at least one extracted lysate, with the serovars Virchow and Heidelberg being the most susceptible to phage infection and the serovars Blockley and Paratyphi being most resistant. Therefore, phage exchange and their transduction events are very common in and between different serovars, and horizontal gene transfer by phage-induced transduction could be an important evolutionary tool for most serovars, with some being more resistant than others to phage transduction. Of the 44 propagated prophage lysates, none could infect *E. coli* or *Y. enterocolitica* two species of the *Enterobacteriacae*, even though *E. coli* shares many features with *Salmonella enterica*, and *Yersinia* phages have been reported as sharing homologies with *Salmonella* phage [34].

## 4. Materials and Methods

### 4.1. Bacterial Isolates

A total of 50 isolates from the *S. enterica* serovar Typhimurium and 50 from the serovar Enteritidis were randomly selected from the strain collection of the Institute of Food Safety, Division of Hygiene and Technology of the University of Veterinary Medicine, Vienna, Austria. Reference strains of the serovar Enteritidis DSM 17420 (German collection of Microorganisms and Cell Cultures, Braunschweig, Germany) and the serovar Typhimurium ATCC 14028 WT (American Type and Culture Collection, Manassas, VA, USA) were used. All the strains and isolates were streaked from stocks at −80 C. The isolates were all serotyped at the Austrian Reference Centre for Salmonellosis (Austrian Agency for Health and Food Safety, Graz, Austria) (see Appendix A for all isolates used).

### 4.2. Prophage Induction

For phage induction, a protocol of Pacifico et al. [35] was used with the following modifications. Isolates were streaked on MH broth (CM0405, Oxoid, Wesel, Germany) and incubated for 24 h at 37 °C in aerobic conditions. One colony was used for overnight culture in liquid broth (LB-broth OXOICM0996B, VWR International, Radnor, TN, USA) under shaking (250 rpm/min) for 24 h at 37 °C. Next, 1 ml of stationary bacterial cells were pelleted at 10,000× *g* and diluted in 2 mL prophage induction buffer (MH-broth 10 mM CaCl_2_; 10 mM MgCl_2_; 2 mM H_2_O_2_), and then incubated for 1 h at 30 °C for phage activation. Lysozyme at a concentration of 200 µg/mL was added and incubated for 1 h at 30 C followed by a centrifugation step at 10,000× *g* for 5 min at room temperature. The supernatant was collected, filtered sterile (0.2 µm-pore filter), treated with DNAase (2U/µL) (95159-01K QuantaBio, Beverly, MA, USA) and used for the infection experiments.

### 4.3. Phage Infection

Phage infection was carried out according to Groisman [21]. The collected lysate for phage infection was used undiluted and in a 1:100 dilution. Overlay plates were incubated at 30 C for 24 h. All the overlay plates were screened for plaque forming by eye and with reflected light microscopy (100–1000×). Plaques were counted on plates containing phage forming units. A total lysate was prepared from these plates, as described in the reference above, using an additional sterile filtering step, DNAse (2U/µL) and RNase (20 mg/mL) (732-3152, VWR International, Radnor, TN, USA) treatment to digest free DNA and RNA.

### 4.4. Analyzing Phage DNA for Antibiotic Resistance Genes

These total lysates were used for DNA preparation using a QIAamp DNA mini kit (Qiagen, Venlo, The Netherlands) and this DNA was used for PCR of resistance genes *sul1*, *sul2*, *tetA*, *tetB*, *tetC*, *bla_OXA_*, *bla_TEM_*, *bla_CMY_*, *strA*, *aadA1*, *dfr1*, *dfr2*, *cmlA* and *cat*, using the primers and methods described in Sacher-Pirklbauer et al. [36]. All 44 lysates from the propagated prophage out of a total of 100 *Salmonella* isolates were analyzed.

### 4.5. Transduction of Antibiotic Resistance

Transduction assays were performed with total lysates positive for any resistance gene, using the method of Shousha et al. [37]. In brief, titration of the bacteriophage was carried out for each phage lysate in tenfold serial dilutions (10^−1^ to 10^−6^) by using plaque counting on overlay plates and expressing phage particles in phage forming units (PFU) per ml. For transduction, a 100 µL overnight culture of either the *S.* serovar Typhimurium ATCC 14028 WT or the *S*. serovar Enteritidis DSM 17420 was used, mixed with 10 µL appropriate phage dilution (10^2^–10^5^) and incubated at 37 °C for 30 min. After adding 2 mL Scholtens’s broth, the mixture was incubated at 37 °C for 2 h shaking. After centrifugation the pellet was diluted in 150 µL modified Scholtens’ broth and plated on MH-plates containing either ampicillin at 15 µg/mL, sulfonamids at 25 µg/mL, streptomycin at 15 µg/mL and chloramphenicol at 30 µg/mL. The plates were incubated at 37 C for 48 h and screened for colonies.

### 4.6. Defining Host Range of Induced Prophage Lysates

The host range of total lysates was screened using a spot lysis assay. In brief, bacterial strains to be tested for lyses were plated on MH-agar and a drop of 5 µL of the undiluted lysate was pipetted onto the agar plate (up to 15 drops per plate). Lysis was determined after appropriate incubation time and temperature (Table 1). In total, 48 *S. enterica* food isolates of different serovars, 50 *E. coli* food isolates and 50 *Yersinia enterocolitica* food isolates were screened for phage sensitivity.

## 5. Conclusions

Certain *Salmonella* serovars are highly resistant to antibiotics with so-called multidrug resistance phenotypes. Therefore, phage therapy could be an effective alternative to antibiotic therapy when invasive clinical signs of disease are prevalent. However, there are few reports of human phage therapy for salmonellosis, but commercialization is close in animal therapy and in the food sector. Nonetheless, reports of the possibility of virulence, toxin and antibiotic resistance gene transduction, and phage evolution (hybrid phage) must be considered before the use of phage application is widely adopted. Here we show that 80% of food and human isolates of the Salmonella enterica serovar Typhimurium contain intact prophage that could be released by oxidative stress, and that these lysates were able to infect a range of different Salmonella serovars. Although no broad host range lysate has been isolated that infects other Enterobacteriaceae, some of these lysates contain antibiotic resistance genes or gene fragments detected by PCR. The difference between the serovars Typhimurium and Enteritidis in the propagation of prophages and the detection of resistance genes is remarkable.

## Figures and Tables

**Figure 1 antibiotics-12-00595-f001:**
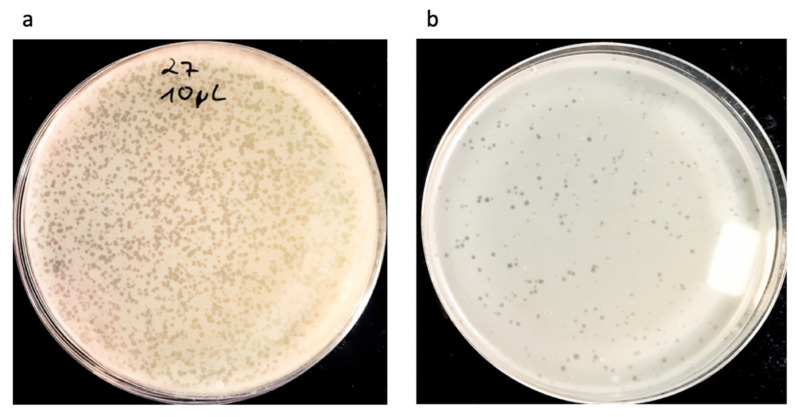
Lysate of propagated prophage of (**a**) PSTy27 a *S*. *enterica* serovar Typhimurium isolate on host strain ATCC 14028 WT and (**b**) PSEn7 a *S*. *enterica* serovar Enteritidis on host strain DSM 17420 (pictures were taken by the authors).

**Figure 2 antibiotics-12-00595-f002:**
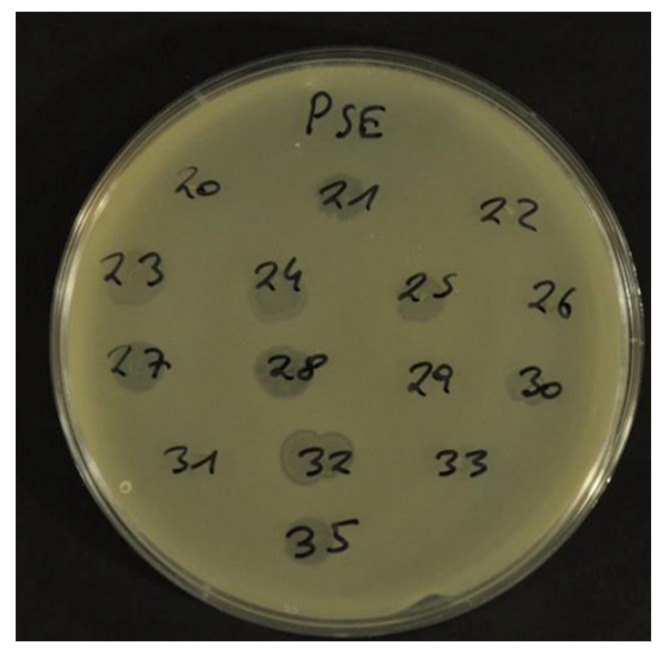
Spot lysis assay on DSM 17420 strain (picture was taken by the authors).

**Table 1 antibiotics-12-00595-t001:** PCR detection of resistance genes in phage DNA isolated from propagated lysates of *S*. serovar Typhimurium.

Lysate	*bla_TEM_*	*cmlA*	*strA*	*sul1*	*sul2*
PS1	−	+	−	−	−
PS2	−	+	−	−	−
PS3	−	+	−	−	−
PS4	−	−	−	+	−
PS9	−	+	−	−	−
PS14	−	−	−	−	+
PS17	−	+	−	−	−
PS19	−	−	−	−	+
PS21	−	−	+	+	+
PS25	−	−	−	+	−
PS26	−	−	−	−	+
PS27	−	−	+	+	+
PS28	−	−	+	−	+
PS33	−	−	+	−	+
PS45	−	+	−	−	−
PS46	−	+	−	−	−
PS47	−	+	−	−	−
PSL18B	−	+	−	+	−
PSL36B	−	+	−	−	−
PSLs157	+	−	−	−	+

**Table 2 antibiotics-12-00595-t002:** Lysis of different *S*. serovars by the propagated prophage lysates.

Serovar/Strain	Number of Lysates Serovar Typhimurium	Number of Lysates from Serovar Enteritidis
ATCC 14028	40	0
DSM 17420	28	4
Blockley	3	0
Heidelberg	34	2
Parathyphi B	5	1
Typhimurium	0	0
Virchow	36	3
Hadar	2	0
Rissen	2	0
Kentucky	2	0
B-Group	6	0

## Data Availability

All data have been described and made available by tables, figures and Appendix A.

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
