# Peer review of "Salmonella Prophages, Their Propagation, Host Specificity and Antimicrobial Resistance Gene Transduction"

_antibiotics, 2023, doi:10.3390/antibiotics12030595_

Round 1

Reviewer 1 Report

Manuscript: antibiotics-2228850

Title: Salmonella Prophage, their Propagation, Host Specificity and Antimicrobial Resistance Gene Transduction

The manuscript is about the phage study where the authors analyze the Salmonella prophage, their propagation, host specificity and antimicrobial resistant gene transduction in several bacterial models. They used Salmonella serovars, Escherichia coli and Yersinia enterocolitica.

Comments and suggestions

Line 51. My first comment is about the meaning of a one word used in the line 51, I don’t know if that is a correct use of the word “morons”. I couldn’t understand the context. Please clarify.

Figure 1. In figure one, I suggest that they have to try explain a little bit more all the figure with a lot of details.

Material and methods

L66-67. Some information isn’t clear; First: you say that the bacteria selected for the experiments not harbor inducible prophage (lines 66-67). Second: What phages did you used to perform these experiments? Third. If the bacteria not contain prophage in their genomes, how is it possible that the inductions occur? All this information is not clear.

You never mention the phages that were used in this study.

Lines 200-202. I reviewed other manuscripts about this topic, and I would like to know why they didn’t count the phage forming units in the supernatant collected. It would be better if you know the phage concentration.

Line 213. There is a misspelled word “decribed”, please change to “described”. Please check all document.

Line 213. Please include the name of the author in this format: “primers and methods described in Sacher-Pirklbauer [35]”.

Line 214. Please indicate the number of total positives lysates.

Results

Lines 62-75. In this section there is something that confuses me, they mention that the phages were induced in bacteria were the prophages were absent.

Table 1. They mention that the PCR detection of the antibiotic resistances genes were possible in propagated lysates. How can they ensure that antibiotic resistance is not already present in the bacterial genome? They had to test for antibiotic resistance before infection with the phages.

Author Response

Dear Reviewer,

Many thanks for your valuable comments and suggestions on Manuscript: antibiotics-2228850

Reply 

Title: Salmonella Prophage, their Propagation, Host Specificity and Antimicrobial Resistance Gene Transduction

The manuscript is about the phage study where the authors analyze the Salmonella prophage, their propagation, host specificity and antimicrobial resistant gene transduction in several bacterial models. They used Salmonella serovars, Escherichia coli and Yersinia enterocolitica.

Comments and suggestions

Line 51. My first comment is about the meaning of a one word used in the line 51, I don’t know if that is a correct use of the word “morons”. I couldn’t understand the context. Please clarify.

Has been clarified in the text with a short explanation of “morons”

Figure 1. In figure one, I suggest that they have to try explain a little bit more all the figure with a lot of details.

A detailed description about the overlay method and the prophage induction has been added in the text line 77-82.

Material and methods

L66-67. Some information isn’t clear; First: you say that the bacteria selected for the experiments not harbor inducible prophage (lines 66-67).

Only the host strains we used did not harbour any prophage which are ATCC 14028 and DSM 17420, which makes them good hosts to be infected by phages isolated from our field isolates (human and food isolates). We tried to make it clear in the text (line 66-70).

Second: What phages did you used to perform these experiments?

The phages used were propagated out of our food and human isolates. We tried to make it clearer in the text (line 64-65).

Third. If the bacteria not contain prophage in their genomes, how is it possible that the inductions occur? All this information is not clear. You never mention the phages that were used in this study.

See above, we rephrased to make it clear to the reader (line 64-74).

Lines 200-202. I reviewed other manuscripts about this topic, and I would like to know why they didn’t count the phage forming units in the supernatant collected. It would be better if you know the phage concentration.

Many thanks for your valuable comment. Phage plaque counts would not be of any additional value as phage induction did induce prophages and these were multiplied when activated. But there is no possibility to assume when this activation happens, which would be related to plaque counts as well as the number of prophages that were activated.

For lysates used in the transduction studies, phages were counted and used in concentrations of 102-5 PFU (plaque forming units) as described in M&M section.

Line 213. There is a misspelled word “decribed”, please change to “described”. Please check all document.

Many thanks –changed.

Line 213. Please include the name of the author in this format: “primers and methods described in Sacher-Pirklbauer [35]”.

Done

Line 214. Please indicate the number of total positives lysates.

Added.

Results

Lines 62-75. In this section there is something that confuses me, they mention that the phages were induced in bacteria were the prophages were absent.

Clarified

Table 1. They mention that the PCR detection of the antibiotic resistances genes were possible in propagated lysates. How can they ensure that antibiotic resistance is not already present in the bacterial genome?

We only tested the phage DNA and not the bacterial DNA. Host strains DSM 17420  and ATCC 14028 do not harbour any resistance genes. We clarified this in the text (line 115-116, 124)

They had to test for antibiotic resistance before infection with the phages.

Genomes and phenotypic resistance in these strains is published and accessible. This has been included in the manuscript now (line 129-131)

Reviewer 2 Report

1. Figure 1 - please provide the source for the images in the description. Is it from laboratory collection or from other research group?

2. As authors point in line 142 - bacteria are unlikely to come into contact with Mitomicin C, however there are antibiotics that can induce prophages. The rewiever suggests the phage induction experiment should be performed using either mytomicin C or other antibiotics that could induce prophages from Gram negative bacteria. It could improve the scientific value of this paper.

3. Materials and methods - please divide this section not into paragraph but give the headings to each method as in current form it is very difficult to read and to understand. I.E. 4.1 Bacterial strains 4.2 Prophage induction protocol 4.3 Phage infectio etc...

4. Have the MIC values been analysed for the strains carrying the antibiotic resistant genes? The presence of the gene does not equal resistance. It is something to be concerned about and should be taken into consideration but the rewiever feels the statement that prophages are responsible for resistance spread in this case is not fully spupported by the presented results. Please provide MIC values for the strains.

5. There is no references to supplementary materials in the text. Furthermore, the reviewer feels that the table with host range of phages should be in the main text and not as supplementary material as the host range of those phages is something that is being discussed but no visual data are given apart from one picture of a double-layered agar plate

6. Have the strains carrying the prophage been checked for superinfection resistance?

Author Response

Dear reviewer,

Thanks for your comments on Manuscript: antibiotics-2228850

  1. Figure 1 - please provide the source for the images in the description. Is it from laboratory collection or from other research group?

These images have been produced in the lab during the studies and have not been published elsewhere. We mentioned this now in brief in the figures’ description.

  1. As authors point in line 142 - bacteria are unlikely to come into contact with Mitomicin C, however there are antibiotics that can induce prophages. The rewiever suggests the phage induction experiment should be performed using either mytomicin C or other antibiotics that could induce prophages from Gram negative bacteria. It could improve the scientific value of this paper.

Previous to this study we also tested E. coli isolates for prophage induction with different agents and procedures as are mitomycin C, H2O2, ciprofloxacin and UV-light. We also tested now additionally four isolates of serovar Typhimurium and serovar Enteritidis each, two with inducible prophage (using H2O2) and two with non inducible prophage with both H2O2 and mitomycin C and added the results (line 86-90). Interestingly, the in one serovar Typimurium isolate prophage were inducible with H2O2 but not with mitomycin C.

  1. Materials and methods - please divide this section not into paragraph but give the headings to each method as in current form it is very difficult to read and to understand. I.E. 4.1 Bacterial strains 4.2 Prophage induction protocol 4.3 Phage infectio etc...

Done.

  1. Have the MIC values been analysed for the strains carrying the antibiotic resistant genes? The presence of the gene does not equal resistance. It is something to be concerned about and should be taken into consideration but the rewiever feels the statement that prophages are responsible for resistance spread in this case is not fully spupported by the presented results. Please provide MIC values for the strains.

As no transduction was possible (no resistance) no colonies on plates with appropriate antibiotics content. We rephrased the text and clarified.

  1. There is no references to supplementary materials in the text. Furthermore, the reviewer feels that the table with host range of phages should be in the main text and not as supplementary material as the host range of those phages is something that is being discussed but no visual data are given apart from one picture of a double-layered agar plate

We added part of the supplemental data on host range in the results section as a table and added a comment to the supplemental data.

  1. Have the strains carrying the prophage been checked for superinfection resistance?

No isolates have been checked for reinfection. We added a sentence in the discussion section on it.

Reviewer 3 Report

Salmonella is a constant health risk to both humans and animals and considering the spread of antibiotic resistance among pathogenic and commensal bacteria, the development of novel approaches to combat such microorganisms is crucial. The manuscript is well written, but the aim of the study is not clear, please add the objectives. However, I would also consider the suggestions listed below:

Line 36: Please check the spelling and correct: occurs, I guess.

Line 38: Consider adding a coma.

Line 46: I would suggest using the plural: phages.

Author Response

Dear reviewer,

Thanks for your review on Manuscript: antibiotics-2228850

Salmonella is a constant health risk to both humans and animals and considering the spread of antibiotic resistance among pathogenic and commensal bacteria, the development of novel approaches to combat such microorganisms is crucial. The manuscript is well written, but the aim of the study is not clear, please add the objectives. However, I would also consider the suggestions listed below:

Line 36: Please check the spelling and correct: occurs, I guess.

Thanks. Corrected.

Line 38: Consider adding a coma.

Done.

Line 46: I would suggest using the plural: phages.

Changed.

Round 2

Reviewer 1 Report

The authors resolved all the comments and suggestions properly.

Reviewer 2 Report

The authors have answered my concerns in rather satisfactiory matter. Please read throughout the text once more for minor spell and grammar check.